# Are Tourists Facilitators of the Movement of Free-Ranging Dogs?

**DOI:** 10.3390/ani12243564

**Published:** 2022-12-16

**Authors:** Elke Schüttler, Jaime E. Jiménez

**Affiliations:** 1Sub-Antarctic Biocultural Conservation Program, Universidad de Magallanes, Teniente Muñoz 166, Puerto Williams 6350000, Chile; 2Cape Horn International Center (CHIC), O’Higgins 310, Puerto Williams 6350000, Chile; 3Department of Biological Sciences and Advanced Environmental Research Institute (AERI), University of North Texas, 1155 Union Circle #305220, Denton, TX 76203, USA

**Keywords:** camera-trapping, *Canis lupus familiaris*, Chile, companion animal, invasive predator, protected area, questionnaire, South America, wildlife management

## Abstract

**Simple Summary:**

Free-ranging dogs are of major conservation concern worldwide as they negatively affect wildlife. This is particularly true for the Global South, where dogs even roam free in and around protected areas. Whether tourists who visit those biodiversity-rich areas play a role in facilitating the access of dogs to nature is largely unknown. Using a combined approach of questionnaires with tourists and camera-traps along trekking trails in the Chilean Cape Horn Biosphere Reserve, this study demonstrates that free-ranging dogs indeed followed tourists—even on several day trips—into protected areas. Although scientists and governmental agencies are aware of the problem and responsible pet ownership strategies are beginning to be implemented, the tourism sector must be explicitly integrated into wildlife conservation management strategies. Awareness campaigns promoting the confinement of dogs should be urgently implemented among tourism operators, hostel owners, and the tourists themselves—not only on behalf of wildlife but also to guarantee the attractiveness of green tourism destinations.

**Abstract:**

Dogs are the most abundant carnivores on earth and, as such, negatively impact wildlife. Free-ranging dogs roam in many protected areas, which in turn are often tourist destinations. Whether tourists influence their roaming is largely unexplored but highly relevant to wildlife conservation. To address this question, we obtained (i) 81 completed questionnaires from tourists on their experience with free-ranging dogs in the remote Cape Horn Biosphere Reserve, Chile, and (ii) photographs of three camera-traps placed next to trekking trails (*n* = 87 trap days). A third of the participants were followed by dogs for up to four days, and 39% saw free-ranging dogs on their hikes, but neither feeding dogs nor fear of them had any influence on whether tourists were followed by dogs. Camera-traps yielded 53 independent dog sequences, recorded 32 individuals plus 14 unidentified dogs, of which only one was leashed, with a frequency of one dog every 28^th^ person. In 17% of 53 sequences, dogs were photographed together with hikers carrying large backpacks for several-day trips. We conclude that tourists are facilitators for the movement of dogs and highlight the importance of the engagement of the tourism sector in wildlife conservation in and close to protected areas.

## 1. Introduction

Free-ranging dogs (*Canis lupus familiaris*) are an underestimated driver of biodiversity loss [1]. Among threatened and extinct vertebrates, dogs most commonly affect mammals, followed by birds, reptiles, and amphibians; they typically prey upon wildlife (79%) but also disturb wildlife, transmit diseases, compete, or hybridize with wildlife (review in [2]). Conservation strategies are challenging as dogs roam free in most societies [3]. Unconfined dogs occur under varying interchangeable conditions [4]. They live in urban, village, or rural environments, may be owned or unowned, and depend to different degrees on human resources or are even independent of them (i.e., feral dogs) [5]. However, not all free-ranging dogs have equal access to wildlife. Rural dogs usually have a wider ranging behavior, and their diet is less human-dependent compared to urban dogs (review in [5]).

Of particular concern are free-ranging dogs in close proximity to natural or protected areas. Studies have found interference competition between free-ranging dogs and Indian foxes (*Vulpes bengalensis*) in a grassland sanctuary in India [6], an intense activity of unconfined owned, unowned, and feral dogs in an Atlantic Forest reserve in Brazil [7], and a reduced occupancy of wild animals under the presence of free-ranging dogs in a network of giant panda (*Ailuropoda melanoleuca*) reserves in China [8]. This is also the case for Chile, where the dog population in relation to the human population is relatively high compared to the world estimate (human-to-dog ratio in Chile 4:1 versus global 7:1 in 2011, [4]) and where the majority of rural dogs are allowed to roam free (88% in [9]; 92% in [10]). Particularly in southern Chile, where more studies are available, free-ranging dogs are reported in and close to protected areas. Here, unconfined dogs negatively affect a variety of vulnerable, endemic mammalian species: Darwin’s foxes (*Pseudalopex fulvipes*) changed their diurnal activity patterns in the presence of dogs [11], southern pudu deer (*Pudu puda*) avoided spatial co-occurrence with dogs [12], and huemul deer (*Hippocamelus bisulcus*) were commonly depredated by dogs [13]. Apparently, dogs enter protected areas in Chile, although their access is prohibited by the administration of the National Forest Corporation (CONAF). On a global scale, dogs are often either excluded from strictly protected areas such as national parks (e.g., in Brazil [14]) or only allowed on a leash (e.g., in Australia [15] and the United States [16]). However, compliance with dog restrictions seems to be rather low in general (reviewed by [17]). While there is an ongoing debate on how regulations might be effectively enforced [18], there are often free-ranging dogs living near and accessing protected areas [19,20]. Dogs whose owners let them out without supervision can move large distances from their homes (e.g., up to 4.5 km in Tanzania [21]; 17 km in Australia [22]; 28.4 km in our study area [23]), and they visit a variety of habitats (e.g., forest in Brazil [24]; pastures in Chile [25], beaches in Mexico [26]). However, what drives them to visit the wilderness is still less clear.

We already have a general understanding of which biological or husbandry-related characteristics influence the roaming behavior of dogs, such as sex [22], age [27], sterilization [28], or the quality of food [26]. But surprisingly, how human interaction with free-ranging dogs, which are domesticated and highly social animals, shapes their movement has been investigated less frequently. Recently, Saavedra-Aracena et al. demonstrated that the dog–owner bond, particularly the exploration behavior in the absence of the owner, explained further roaming [23]. Research on the human–dog link determined that free-ranging dogs tend to follow persons other than their owners [29,30]. Therefore, might tourists, who target green destinations, specifically attract dogs to nature? To our knowledge, this question has not been quantitatively addressed, but there is anecdotal evidence. In southern Chile, Silva-Rodríguez & Sieving [12] mentioned tourists who brought dogs to protected areas harboring endangered pudus, and Navedo et al. [31] observed a tourist with dogs next to migratory shorebirds on Chiloé Island. Furthermore, in some protected areas in Brazil, visitors appear to appreciate stray dogs as guides and for companionship [19].

Thus, a better understanding of the role of tourists as voluntary or involuntary drivers of dog movement is crucial for the design of management strategies, particularly near wilderness areas that lack regulations—or their enforcement—on dog restrictions. In this study, we asked whether tourists can facilitate the entrance of dogs into natural areas. Our study was motivated by recent data on GPS-collared dogs accompanying tourists on several-day trekking trips in our study area [32]. We (i) assessed the experience (i.e., sightings, perceptions) of tourists with free-ranging dogs, (ii) evaluated which factors (i.e., feeding, tourists’ fear of dogs) had an effect on whether dogs followed tourists, and (iii) collected photographic evidence on dogs (i.e., numbers per person, per trip length) accompanying tourists on their hikes into remote sub-Antarctic ecosystems in the southernmost biosphere reserve of the world. We close our study with management implications for enhancing biodiversity conservation in Chile´s protected areas and beyond.

## 2. Materials and Methods

### 2.1. Ethics Statement

Each questionnaire started with an informed consent page explaining the affiliation and aims of the study, funding information, anonymous data storage, and information on the dissemination of the results. This page was signed by the participants.

### 2.2. Study Area

Our study took place on Navarino Island, southernmost Chile (55° S, 67° W, Figure 1). The island (2528 km^2^) is part of the Cape Horn Biosphere Reserve (19,172 km^2^ terrestrial surface) within the Magellanic sub-Antarctic ecoregion [33], still characterized by a low human impact with respect to human population and livestock densities, forest change, land cover, and nighttime lights [34]. The biosphere reserve has a density of only 0.1 persons per km^2^ (2063 inhabitants, national census in 2017), and common landscapes are southern beech forests (*Nothofagus* spp.), peatbogs, Andean habitats, fjords, and lakes (photographs in Appendix A). The native vertebrate fauna is rich in birds (>150 spp. [35]) but scarce in mammals and fish (≤10 species [36]); amphibians and reptiles are completely lacking. A major conservation challenge in the region is invasive species, particularly mammalian predators such as American mink (*Neovison vison*), as well as free-ranging cats (*Felis catus*) and dogs, as they occupy a new niche of terrestrial predators on some islands [36,37,38]. Chile lacks a framework for UNESCO biosphere reserves as a legal figure in the national protected area system except for core zones, which are often national parks; therefore, on Navarino Island, dogs are not prohibited (but must be confined on leashes or within fenced yards).

Puerto Williams, located on Navarino Island, is the only major town in the biosphere reserve, where the majority of the inhabitants of the reserve live. Additionally, Puerto Toro, a small fishing town harboring a couple of families, exists on the eastern coast of the island. The terrestrial infrastructure is limited to a dirt road along the northern coast of the island, which gives access to eight permanent farms. Economic activities are focused on defense and public services, construction, commerce, small-scale livestock, artisanal fisheries, and tourism. The archipelago is an attractive national and international tourist destination. Yearly, between 2010 and 2020, a mean of 7100 ± 2840 tourists visited the Cape Horn National Park, which is the core zone of the Cape Horn Biosphere Reserve [39]. Moreover, 840 ± 490 tourists visited the natural attractions of Navarino Island, which include trekking trails famous among alpinists (register of the local police station). Furthermore, the area recently has been declared 1 of 35 national zones of tourist interest [40], aiming at fostering tourism. With the construction of new maritime infrastructure and roads, together with land subdivisions and settling of formerly uninhabited rural areas of Navarino Island, the number of tourists and residents will substantially raise in the near future. Likely, this has consequences for the movement of free-ranging dogs into wilderness areas.

### 2.3. Questionnaire Survey

Between December 2015 and April 2016, 490 questionnaires were administered in three languages (Spanish, English, German). We distributed the questionnaires among the active hostels (13/13) and tourism operators (2/4) in Puerto Williams. Approximately every two weeks, we visited the tourism establishments to collect completed questionnaires and restock them. The questionnaire was composed of 14 questions (Appendix A). Seven questions regarded the tourist´s experience with dogs (i.e., town dogs following them when hiking, observations of dog interactions with other animals, feeding of dogs, sightings of dogs outside the village, level of fear when encountering free-ranging dogs, problems experienced with dogs, other comments), and seven questions about the personal background of the tourist (i.e., gender, age, nationality, highest completed education level, occupation, time spent in Puerto Williams up to date, and reason for visit). The questionnaire also contained two maps of Navarino Island showing the trekking trails at different scales, where locations of dog sightings were drawn.

### 2.4. Camera-Traps

During an operating period of one month (25 January 2019–22 February 2019), three camera-traps (model Bushnell Trophy Cam) in total were placed at a distance of 110–140 m from the starting point of three trekking trails each (see also [41]), yielding 87 camera trap-days. We attached each camera to a tree (height ~0.5 m) at ~10 m perpendicular to the trail and ensured the movement sensor was activated by dogs or persons walking on the trail. Each camera was programmed to take three photographs per trigger every second, with a resolution of 8 MP. We acquired permits for the installation of camera-traps from the regional public property authority.

### 2.5. Data Analysis

We describe the questionnaire data using descriptive statistics, mainly percentages, with differing sample sizes due to missing data in the questionnaires. We compared independent samples of questionnaires using Fisher´s exact, Mann-Whitney, and Wilcoxon rank sum tests with continuity correction with a *p*-level of 0.05. We categorized camera-trap data into (i) dogs accompanying people, (ii) people alone, and (iii) dogs alone. When dogs accompanied people, we distinguished between people with small (for a day-trip) and large (trekking for several days) backpacks. From the photographs of the camera-traps, we extracted information on dogs on the trails (i.e., individual identification, when alone in the photograph, minimum time to person behind or ahead, daytime, number of dogs in groups, wearing of dog collar, leashed/unleashed, direction of movement), as well as the number of hikers. To respect their privacy, we committed to not sharing photographs with any third party nor extracting data other than the abovementioned information in the context of this study. We pixeled the faces of the persons appearing in Figure 4 to guarantee their anonymity. All analyses were performed using program R [42].

## 3. Results

### 3.1. Questionnaire Survey

We obtained 81 questionnaires from eight hostels and two tourism operators (16.5% return rate) completed by non-residents between December 2015 and April 2016. To understand how representative our sample size is, we used the number of tourists registered at the local police station on the three trekking trails during summer 2014–2015 (*n* = 306 tourists). Following Cochran´s formula applying the finite population correction factor for small populations [43], 81 returned questionnaires correspond to a precision level of 9% at a 95% confidence level. The participants were nearly even in gender (51.9% males, 48.1% females), their age was 37.2 ± 13.2 years (mean ± SD; range 18–68), the majority had or were currently acquiring a university education level (85.2%), 9.9 % were technically educated or in technical training, and 4.9% had finished high school. Most participants were either South Americans (45.8%, 42% Chileans) or Europeans (44.4%), but tourists were also from North America (4.9%), the Middle East (3.7%), and Oceania (1.2%). The great majority (86.4%) came for tourism, and 13.6% visited the island for work. Among those who came for tourism, over half (55.7%) explicitly mentioned trekking as their reason for travel. Finally, the mean duration of their stay was 7.1 ± 5.8 days (range 1–30 days, *n* = 80).

A third of the participants (32.9%, 26/79) reported that during their hikes or trekking tours, dogs followed them. In most cases (71.4%, 15/21), it was only one dog, but there was a range of up to three dogs (1.4 ± 0.7 dogs). Dogs followed the tourists between three minutes and up to four days (13.2 ± 28.6 h, median = 1 h, *n* = 20). Most participants thought dogs followed them for company or affection (33.3% of statements, 7/21), followed by curiosity (28.6%, 6/21), and interest in food (23.8%, 5/21); three participants provided hunting, taking care of their territory, and physical exercise as reasons. A considerable number of participants (39%, 30/77) also saw free-ranging dogs outside town. Dogs were seen alone (10 participants) and in groups of up to 10 individuals (2.6 ± 2.2, median 2 dogs, *n* = 27). Dogs were seen on the official trekking trails but also in other areas of the island up to 35.2 km away from Puerto Williams (Figure 1).

Sixteen participants (20.3%, 16/79) saw dogs chasing other animals, among them mainly other dogs (53.3% of statements, 8/15) and birds (20%, 3/15) but also beavers (*Castor canadensis*), American mink, and horses (*Equus caballus*, 4 statements). Most participants reported they did not feed dogs during their stay on the island (84%, 68/81), 10 participants fed dogs sometimes (12.3%, 10/81), and only three on a daily basis (3.7%, 3/81). There was no significant relationship between participants having fed dogs always/sometimes and having been followed by dogs (1.6%, versus 31.7% having been followed without feeding, Fisher´s exact test, *n* = 63, *p* = 0.25).

On a scale from 0 to 10, with 0 meaning no fear and 10 meaning a high level of fear, participants were only moderately afraid of encountering free-ranging dogs outside town (2.7 ± 3.1, median 2, *n* = 78), and 30 participants had no fear at all (38.5%, 30/78). Participants from Europe had a significantly lower level of fear (1.5 ± 2.3, median = 0, *n* = 35) in comparison to participants from South America (4.3 ± 3.3, median = 4, *n* = 35) (Mann–Whitney test, U = 924.5, *n*_1_ = *n*_2_ = 35, *p* = 0.0002), but the level of fear had no significant influence on whether tourists were followed or not by dogs (Wilcoxon rank sum test, Z = 390, *n*_1_ = 21, *n*_2_ = 42, *p* = 0.45). Overall, participants had no problems with free-ranging dogs during their stay on the island (90.4%, 66/73). Those participants who experienced a problem reported dogs either ran after them, barked, or demonstrated aggressive behavior. Figure 2 illustrates the principal findings of the questionnaires.

Finally, the participants had a chance to leave a general comment, which allowed us to better understand how they perceived free-ranging dogs. In total, 40 participants commented on 49 aspects. Almost half of these (42.8%, 21/49) expressed concern regarding free-ranging dogs: eight participants mentioned the need to control the situation or the abandonment of dogs: “It is very important that no animal introduced by humans contaminates the reserve.” (Participant [P] 29), or “…hopefully something can be done with the dogs, stimulate their adoption.” (P51). Five participants stated dogs either interacted with other animals (other dogs or horses) or may have prevented them from observing wild species (birds and beavers): “We would have loved to see beavers during our trekking, but probably due to the dog this was impossible.” (P74). Aggressive dogs were also an aspect of which some participants were concerned: “I tend to be scared of stray dogs.” (P15). Four participants referred to the poor health status of some dogs: “I saw a dog with three legs in the village.” (P49) or “For me it was quite difficult to not feed the dogs. Some of them weren’t in good condition, some had some signs of illness…” (P18). Another frequent series of comments (28.6% of all aspects, 14/49) was that although dogs had not followed them personally, they witnessed dogs following other tourists on trails or reported to have seen dog tracks on the trails. Notably, the rest of the comments (28.6%, 14/49) dealt with the opinion that free-ranging dogs were not a major problem. Five participants even indicated that dogs were friendly and “…seem very happy and healthy.” (P57).

### 3.2. Camera-Traps

During 87 camera-trap days, we recorded 53 independent dog sequences on the three trekking trails; a sequence meaning consecutive photographs of the same individuals at the same site. This totaled 68 dog records (some individuals were photographed repeatedly), from which we identified 32 different dogs. Another 14 dogs remained unidentified individuals, as photographs only showed parts of the animal; hence, we cannot be sure they were new individuals. The flux of dogs per person on all trails considering both directions of the trail was one dog for every 28th person. Dogs were photographed more than every second day during the trapping period (19/28 days). In 20.8% of the sequences (11/53), dogs were recorded in groups of 2–4 individuals (mean 1.3 ± 0.6, median = 1). Forty percent of dogs from which the neck was visible were collared (12/30). In the majority of the sequences (69.8%, 37/53), dogs appeared together with hikers in the photographs or within three minutes of the time lapse (20.8%, 11/53); thus, they were classified as dogs accompanying people (time series in Figure 3). We speculate most hikers were tourists, not owners, particularly those with large backpacks, as we placed camera-traps during the high season of tourism and trekking is the major tourist attraction of the island. Two dogs were separated by 10 and 17 min from photographed hikers, and seven dogs (two groups of three dogs each, one dog alone) were considered free-roaming on their own as time separation between them and hikers was between 2 and 6 h. All dogs were recorded in daylight, but during two sequences, dogs were photographed at night (one dog accompanying hikers at 22:24 h, and a group of three free-roaming dogs at 2:19 h). Only one of all dogs was on a leash.

Dogs accompanied hikers with large backpacks in 9 of 53 sequences (17%). In these cases, only one dog came back later that day together with other hikers and one dog was photographed five days later ascending the trail again with another group of hikers. The other dogs were not recorded coming back on their own during the next days; we suspect they accompanied the hikers during their several-day trips (or went back off-trail). Of the 23 identified dogs accompanying people, only eight dogs were photographed bidirectionally on the same day and none on later days. Finally, five dogs were repeatedly photographed within 2–5 different days, two dogs on two different trails and one dog on all three trails. This latter dog was owned by a local resident family, which we were able to identify. Figure 4 shows photographs of free-ranging dogs together with hikers with small and large backpacks and roaming on their own.

## 4. Discussion

Increasing scientific evidence of the impacts of dogs on wildlife [2,44,45] urgently requires us to better understand the drivers of roaming dogs. In this study, we addressed the role of tourists as facilitators of the access of free-ranging dogs into wilderness areas. Indeed, dogs followed tourists on their hikes, as stated by one-third of respondents (26/79). Further, tourists observed dogs following other tourists (14/81), and camera-traps revealed photographs of dogs together with hikers (1 dog per 28 persons, *n* = 87 camera-trap days). The number of identified dogs from camera-traps (*n* = 32) corresponded to approximately one-quarter (22.7–25.4%) of the total population of free-ranging dogs in Puerto Williams following a photographic capture-recapture survey (126-141 individuals [32]). Dogs even accompanied hikers for several days (15% of respondents, 3/20) and were photographed together with hikers carrying large backpacks (17% of sequences, 9/53). Assuming tourists did not plan to hike with free-ranging dogs, they likely did not carry extra food for dogs, nor were dogs under voice and sight control. Of course, this might be different if hikers were residents taking their own dogs with them. Although we cannot completely discard this possibility, we did not recognize residents carrying large backpacks. However, even leashed dogs walked by their owners have been shown to displace birds in Australia [46], but refer to [16]. Importantly, 65.2% (15/23) of the identified dogs accompanying people were not detected on the same or later days on any of the three trails, indicating the camera-traps did not detect them or dogs came back off-trail.

Why would dogs follow unfamiliar persons? In our study, participants primarily thought company or affection motivated dogs to follow tourists, more so than providing food (33% versus 24%, *n* = 21 statements). This might be a plausible explanation; most tourists reported not having fed dogs (84%, *n* = 81), and we could not detect a significant relationship between tourists feeding dogs and tourists being followed by dogs. Further, the local dog population had relatively high husbandry standards, with 78% of 215 households feeding quality food (i.e., commercial dog food and/or meat [37]). Furthermore, experiments with street dogs in India indicated social rewards, such as petting from unfamiliar humans, can be more important than food if occurring repeatedly [47]. Therefore, we propose that free-ranging dogs may have a deficiency in positive social interactions with their owners that determine aspects of their movement behavior. Alie et al. [48] suggest keepers of free-ranging dogs in Dominica were passive caregivers who did not play, train, or discipline their pets frequently. Similarly, during experiments on the dog–owner bond in our study area (Strange Situation Procedure, [49]), only 5/39 dogs played upon invitation [23]. Future studies are needed that experimentally address what factors influence dogs to follow unfamiliar persons, e.g., type and duration of dog–owner interactions, immediate cues such as food or petting by strangers (e.g., [47]), and inherent factors such as sociability [50]. A better understanding of those effects is not only relevant for wildlife conservation but also for the well-being of dogs.

Furthermore, the influence of group formation in free-ranging dogs warrants attention, as group formation might determine their movements. In our study, tourists were followed on average by 1.4 dogs, observed dogs in groups of 2.6 animals, and were photographed on trails together with 1.3 dogs. Free-ranging dogs often occur in groups of up to four individuals (in Chile [51], China [52], North America [53], Zimbabwe [54]), which allows them to perform pack hunting, targeting even large mammals (e.g., [55]). Among the factors determining the propensity of group formation in free-ranging dogs is the presence of garbage sites [56], but more studies are needed, e.g., examining whether dogs of the same household or more social dogs roam in groups. The number of dogs per household indeed influences their confinement, i.e., multiple dogs in the household tend towards less probable confinement [57].

Finally, most tourists in our study perceived dogs negatively in some aspects. Only one-third (29.6%, 42/81) of the participants did not experience any fear (fear level = 0), did not report any problems, or did not leave a negative comment. In contrast, the negative aspects mentioned by tourists (21 of 49 comments) referred to a wide array of recognized problems of free-ranging dogs, i.e., overpopulation, abandonment, health status, interactions with other animals, lost experience of wildlife observation, and fear. Indeed, due to their different cultural backgrounds, residents and visitors can be at odds over what exactly dog ownership and care mean. In Samoa, for example, free-ranging dogs worsened the holiday experience of tourists coming from Australia and New Zealand [58]. In the Bahamas, North American tourists pitied dogs rather than feeling threatened, but overall perceived roaming dogs as a problem [59]. This finding is similar to what tourists in our study area felt; moderate fear felt by European tourists (1.5 ± 2.3, scale 1–10) versus higher levels of fear by South American tourists (4.3 ± 3.3). Ruiz-Izaguirre et al. [60] also highlighted the different perceptions between Mexican and North American/European tourists towards the welfare of roaming dogs in Mexico. Those culturally related differences in perceptions can influence the overall holiday experience with possible negative consequences for the future development of the tourism sector [59]. Following Bessa et al. [61], fully recognizing the costs of pet impacts on ecotourism in the scientific literature is a pending task.

## 5. Conclusions

To improve policy on responsible dog ownership, it must be acknowledged that tourists can facilitate the access of dogs to nature. Hence, to reduce pet-induced impacts on green tourism destinations, a cross-sectional approach is needed (see also [61]). Several entities should work together to strengthen dog confinement, including the protected area administration, municipalities, veterinarians, and importantly, also the tourism sector. In particular, we recommend (i) improving compliance with the law with regard to dog restriction (culturally sensitive [48,57]), (ii) encouraging an active dog–owner relationship, including common outdoor activities (see also [23,62]), (iii) taking into consideration the overall negative perception of tourists of free-ranging dogs (see also [59]), and (iv) enhancing the education of tourists with regard to their behavior towards free-ranging dogs (a priori, not using them as guides, not feeding them, not petting them). These recommendations should be an integrated component of the sustainable tourism strategy in Chile and beyond, wherever dogs, tourists, and nature meet.

## Figures and Tables

**Figure 1 animals-12-03564-f001:**
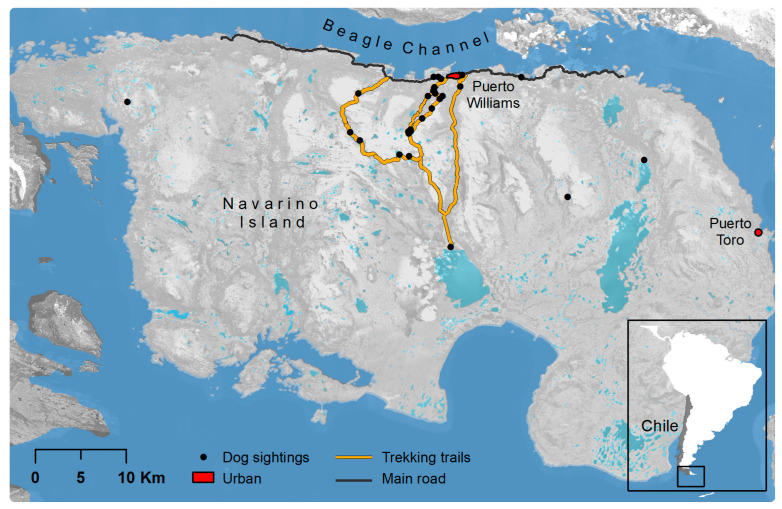
Study area of Navarino Island, Cape Horn Biosphere Reserve, southern Chile. The map highlights the near-pristine character of the island (the only settlements are Puerto Williams and a small fishing town, Puerto Toro) and includes sightings of 58 dogs by 31 tourists (tourist season 2015–2016).

**Figure 2 animals-12-03564-f002:**
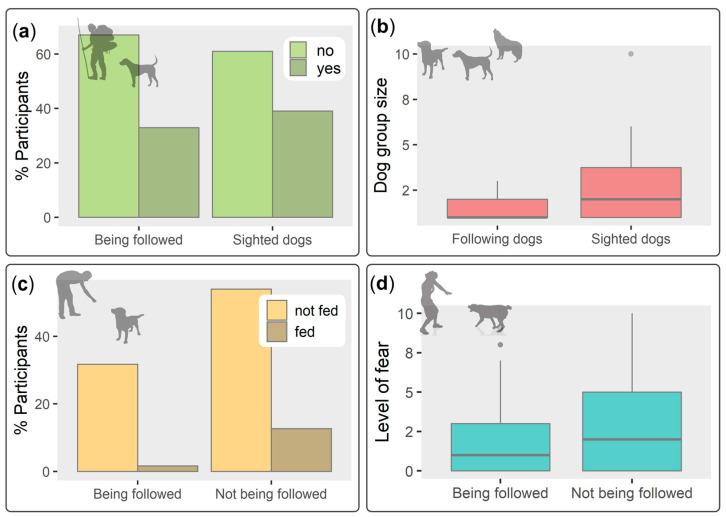
Principal questionnaire results of 81 tourists on free-ranging dogs in southern Chile during summer 2015/2016: (**a**) Participants being followed by dogs (*n* = 79) and having seen free-ranging dogs outside town (*n* = 77); (**b**) group size of dogs having followed participants (*n* = 21) and group size of sighted free-ranging dogs (*n* = 27); (**c**) participants (*n* = 63) being followed by dogs with respect to having them fed or not, and (**d**) participants (*n* = 76) being followed by dogs with respect to their level of fear of free-ranging dogs (0 = no fear, 10 = high level of fear). Illustrations downloaded from Pixabay.com.

**Figure 3 animals-12-03564-f003:**
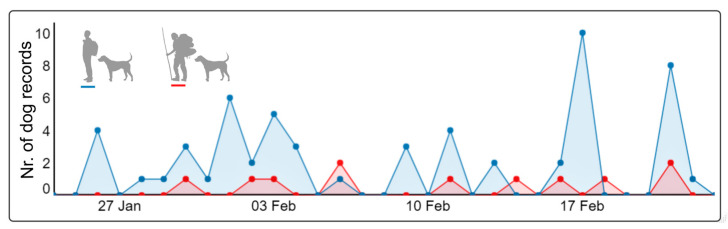
Number of dog records of dogs accompanying hikers with small (blue line) and large (red line) backpacks along three trekking trails in southern Chile (*n* = 56). Data from three camera-traps during an exposure period of four weeks (01/25/2019–02/22/2019). Illustrations downloaded from Pixabay.com.

**Figure 4 animals-12-03564-f004:**
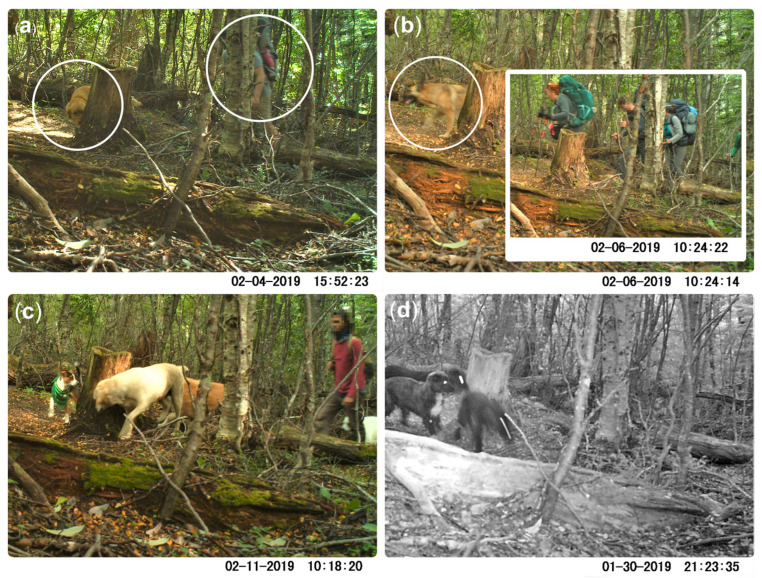
Photographs (cutout of relevant section) of camera-trap in Trail No. 1 in southern Chile during summer 2019. (**a**) Free-ranging dog and tourist using large backpack; (**b**) free-ranging dog photographed 8 s before a group of four tourists using large backpacks (inset); (**c**) four free-ranging dogs with tourist on daily excursion, and (**d**) three free-ranging unaccompanied dogs photographed 1:56 h after last person and 11:51 h before next person.

## Data Availability

For reasons of anonymity, the photographs taken by the camera-traps on trekking trails cannot be made publicly available.

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
