# Peer review of "Are Tourists Facilitators of the Movement of Free-Ranging Dogs?"

_animals, 2022, doi:10.3390/ani12243564_

Round 1

Reviewer 1 Report

Authors should change title and the use of "vector" as it is not appropriate, better if they use the words “facilitators”  Vectors mean arthropods that infect a susceptible host from an infected one.  In this study, dogs do not settle and reproduce in the destiny of the tourist.  In other words dogs did not infect (reproduce) in the wildness because of tourist. 

 Conclusions: the first paragraphs seem out of context.  

An evaluation of biodiversity conservation funding, both from international donors 354 and domestic governments, assigned Chile rank Nr. 9 among the 40 most highly under- 355 funded countries, with Iraq, Djibouti, Angola, Kyrgyzstan, Guyana, the Solomon Islands, 356 Malaysia, and Eritrea on ranks 1-8 [66].

Suggestion: to eliminate. 

Reviewer 2 Report

The manuscript considered very interesting questions - are people help dogs penetrate protected areas. There is a lack of research on that topic, so my expectations for the manuscript were high. I think that the research methods were planned correctly, but I have some comments on the manuscript.

The introduction is written well, but sometimes the whole sentences are too long and hard to follow (also in other parts of ms). Moreover, I’m not sure why the Authors mentioned ‘all dog categories’, indicating that many of them roam free or are free-ranging. As a result, every time I see some of those categories I wondered are those dogs free or not. It also made me wonder which parts of the introduction considered free-ranging dogs. Please specify, the detailed instructions I added to the manuscript.

The detailed comments on methods I added to the manuscript.

The Authors very detailed described the results from questionnaires, but they didn't show enough data from CT. I suggest writing detailed aims of the research and showing analyses for those aims. So far the aims are very general and they rather don't answer the question asked in the title of ms. To be honest, I don't know what was the most important result of the research. I encourage Authors to add graph or graphs to CT results. It might make it easier to understand.

In discussion, the Authors focused on part of the results, not all of them. I didn't find explanations for the results they got and some of paragraphs should be rather moved to the introduction.

Conclusions are not based on results and discussion.

All detailed comments I added to the manuscript.

But I want to mention that the Authors should remove all spaces between the value of the percentage and ‘%’ symbol. I did not highlight them in all ms.

Although I don’t feel qualified to judge the English language, I suggest consulting the language used in the manuscript.

In its present form, the manuscript shouldn’t be published in the journal. I suggest re-writing and resubmission.

I wish the Authors all the best!

Round 2

Reviewer 2 Report

Thank you very much for the opportunity to re-revision the manuscript entitled “Are tourists vectors of the movement of free-ranging dogs?”

I appreciate that the Authors changed the manuscript according to suggestions and thank for all detailed responses and explanations listed in the response letter.

I wonder if “Are tourists involuntary vectors of the movement of free-living dogs?” wouldn’t be a better title for the manuscript. But I want to let the Authors decide about it. Moreover, I’m glad that the Authors added a paragraph about motivation in this manuscript. It’s really good to add such a paragraph in the manuscript, as it’s summarizing and concluding the Introduction.

I still have one concern about the Introduction chapter – In lines, 109-111 (second version of manuscript) the Authors wrote that ‘We (ii) evaluated which factors (i.e., feeding, fear) had an effect on whether dogs followed tourists (…)’ – that suggests that dogs are afraid of tourists, as it is shown in results, and that wasn’t evaluated.

I appreciate all changes that the Authors made in the Methods chapter.

I think that changes made to the map were necessary. The map is now very compatible with study area descriptions (it is shown where the urban area is located), which also improves the manuscript and makes it more clear. The Authors also added some important details to the methods chapter – for example, the height of camera traps. I usually add the number of camera trap days in the methods and materials chapter (as it is concerned rather as material, without distinguished number of photos with focal species), as it was done for number of distributed questionnaires. Nevertheless, if the Authors want to keep this detail in the results, I might understand their point of view.

Results. I still think that writing a percentage and the numbers in parenthesis is not necessary (I would rather write the percentage and number of questionnaires - for example, 33 %, n = 79). But according to the literature provided by the Authors, I agree that showed way is correct. I appreciate all changes provided by the Authors in this section of the manuscript, also in the graph (especially Figure 2c).

Summarizing the motivations and methods, the Authors:

(i) assessed the experience (i.e., sightings, perceptions) of tourists with free-ranging dogs,

(ii) evaluated which factors (i.e., feeding, fear) had an effect on whether dogs followed tourists,

and (iii) collected photographic evidence on dogs (i.e., numbers per person, per trip length) accompanying tourists on their hikes;

Mind above, I don’t understand the result presented in Figure 2d. Again, please specify in the goals/motivations that you mean the tourists’ fear of dogs.

One question to the following part of the response letter:

“Line 242: The reviewer asks whether one sequence also registered more than one dog and that we possibly recorded 68 individuals.

Response: That is true, one sequence could also record a group of dogs. We noticed that we had to clarify the first lines of this paragraph. We corrected in: “During 87 camera-trap days we recorded 53 independent dog sequences, i.e., consecutive photographs of the same individuals at the same site, on the three trekking trails. This totaled 68 dog records (some individuals being photographed repeatedly), from which we identified 32 different dogs. Another 14 dogs remained unidentified individuals.” Does it mean that the Authors recorded 46 individuals? I don’t understand what it means “Another 14 dogs” – why another? Please, explain that.

I appreciate all changes that the Authors made in the Results chapter. The results chapter I found now well-written. Especially new graph is what was missing before.

The Authors rewrote some parts of the discussion, so the structure of the discussion is clearer now. They revised which results of the questionnaire were underrepresented and added the respective sections (this was the case for some results, particularly from the general comments of the questionnaire, on the perception of free-ranging dogs). I think that the discussion is well-written now. The Authors also improved the Conclusions.

To conclude, I want to thank the Authors for all their detailed explanations and changes made in the manuscript. The submitted manuscript is very interesting, and in its present form is well-written and easy to follow. I want to congratulate the Authors on their hard work in improving the manuscript. Besides three comments I don’t have any other future remarks. I recommend publishing the manuscript, after minor revision. I wish the Authors all the best.
